# Using TLS-Measured Tree Attributes to Estimate Aboveground Biomass in Small Black Spruce Trees

**Steven Wagers** [1,2,*], **Guillermo Castilla** [2], **Michelle Filiatrault** [2] and **G. Arturo Sanchez-Azofeifa** [1]

1   Centre for Earth Observation Sciences (CEOS), Department of Earth and Atmospheric Sciences, University of Alberta, Edmonton, AB T6G 2E3, Canada; arturo.sanchez@ualberta.ca
2   Canadian Forest Service, Natural Resources Canada, 5320 122 Street Northwest, Edmonton, AB T6H 3S5, Canada; guillermo.castilla@nrcan-rncan.gc.ca (G.C.); michelle.filiatrault@nrcan-rncan.gc.ca (M.F.)
*   Correspondence: steven.wagers@nrcan-rncan.gc.ca

**Abstract:** *Research Highlights*: This study advances the effort to accurately estimate the biomass of trees in peatlands, which cover 13% of Canada's land surface. *Background and Objectives:* Trees remove carbon from the atmosphere and store it as biomass. Terrestrial laser scanning (TLS) has become a useful tool for modelling forest structure and estimating the above ground biomass (AGB) of trees. Allometric equations are often used to estimate individual tree AGB as a function of height and diameter at breast height (DBH), but these variables can often be laborious to measure using traditional methods. The main objective of this study was to develop allometric equations using TLS-measured variables and compare their accuracy with that of other widely used equations that rely on DBH. *Materials and Methods*: The study focusses on small black spruce trees (<5 m) located in peatland ecosystems of the Taiga Plains Ecozone in the Northwest Territories, Canada. Black spruce growing in peatlands are often stunted when compared to upland black spruce and having models specific to them would allow for more precise biomass estimates. One hundred small trees were destructively sampled from 10 plots and the dry weight of each tree was measured in the lab. With this reference data, we fitted biomass models specific to peatland black spruce using DBH, crown diameter, crown area, height, tree volume, and bounding box volume as predictors. *Results:* Our best models had crown size and height as predictors and outperformed established AGB equations that rely on DBH. *Conclusions:* Our equations are based on predictors that can be measured from above, and therefore they may enable the plotless creation of accurate biomass reference data for a prominent tree species in a common ecosystem (treed peatlands) in North America's boreal.

**Keywords:** terrestrial laser scanning; biomass; black spruce; allometric equations




## 1. Introduction

Forests play a major role in the carbon cycle, as they are some of the most important carbon sinks on Earth [1]. Trees absorb carbon dioxide through the process of photosynthesis [2], removing carbon from the atmosphere and storing it as biomass [3]. Carbon has a major effect on the climate system [4], so being able to accurately estimate the amount of carbon being stored in forest ecosystems is necessary for good climate models. It has been estimated that roughly half of the biomass of a tree is carbon [5–7]. The boreal region in Canada (550 million hectares, of which 270 million hectares is forest) [8] is estimated to contain more than 200 billion tons of carbon [9,10].

Estimating the above ground biomass (AGB, the dry weight of trees excluding their roots) of trees often involves the use of allometric equations that rely on other tree attributes as predictors [5,6,11–13]. Calibration of these equations is often expensive as it requires processing and weighing dozens if not hundreds of harvested trees, but once the calibration processes has been completed, one can use the tree attribute(s) with the calibrated equations to obtain AGB estimates. In Canada, large country-wide efforts have taken place to develop

AGB allometric equations, such as those put forth by Lambert et al. [14] which were later updated by Ung et al. [15]. These equations are used by the Canadian National Forest Inventory for individual tree AGB estimation and they rely on height and diameter at breast height (DBH) [15], which are commonly used for similar AGB equations in other areas of the world [6,11,16]. However, these equations are typically calibrated using commercially sized trees and may not work as well for small trees.

In recent years, measurements of tree parameters forest inventory plots have been conducted effectively by using technologies such as terrestrial laser scanners (TLS) [17] and airborne laser scanners (ALS) [18]. ALS, which gained popularity in the 2000s for its ability to cover large areas and measure plot level attributes, has become an essential tool for forest inventories in many countries [19]. One drawback is that most ALS data have point densities between 1 and 10 pts/m$^2$ [20], limiting their ability to be used for individual tree parameters that require more detail. TLS on the other hand produces much denser datasets, but it can only be used for smaller areas such as forest inventory plots, and each plot requires several scans to tackle occlusion, making this technique time consuming [21]. However, TLS produces highly detailed, 3-dimensional (3D) point clouds [6] that can be used to locate trees within the plot and measure a number of their attributes with high accuracy [22–24]. These point clouds consist of a myriad of points each corresponding to a recorded return from a laser pulse emanating from the scanner that hits the surface of an object. These point clouds also enable researchers to virtually return to the plot at any time to better interpret field data or check for errors and outliers. Although the use of TLS has many benefits, there are some drawbacks as well. Scanning equipment can be expensive and heavy, and it can be time consuming to conduct the scans [25]. Point clouds can sometimes contain data gaps due to occlusion, particularly in dense forests where branches and stems block the laser beam emitted by the scanner from reaching anything behind them [22]. This occlusion is one of the biggest concerns when using TLS, often compromising the data's usefulness [22]. It can, however, be partially reduced by setting up properly planned scan stations at multiple positions in and around the plot [22,26] or, when considering occlusion on leaf area density, by using some interpolation process such as kriging that creates estimators on the basis of spatial information for regions of the point clouds where data do not exist [27]. An example of how occlusion can affect data can be seen when measuring DBH. While the DBH of large trees has been accurately measured using TLS in many cases [28–30], when parts of the tree stems are occluded, TLS-based DBH measurements become less accurate [31].

In the past, AGB has often been estimated using DBH and in fact many allometric equations have been developed for different tree species around the world using DBH [14,15,32–35]. However, DBH cannot be reliably measured from above the tree canopy using ALS or even DLS (drone laser scanning, a subtype of ALS undertaken from low flying drones carrying a small LiDAR instrument). Hence, for the purposes of estimating the AGB of individual trees with these technologies, other allometric equations that do not include DBH are more attractive, such as those that rely on crown attributes. Recent studies have shown the potential of crown parameters, such as crown diameter, to accurately estimate AGB [13,36]. Crown parameters also have the important advantage of being measurable using ALS (assuming high enough point density), whereas DBH can be very difficult to measure from above, and requires higher point densities than those in typical ALS point clouds [37]. ALS can cover large areas of land quickly [38], making models that use ALS-obtainable parameters as predictors very attractive for obtaining extensive AGB reference datasets without relying on ground plots. This could have implications on satellite mapping of AGB as well, particularly for missions such as ICESat-2 [39] and GEDI (for areas below 51° N) [40]. Recent studies have tested methods for estimating biomass with data from these missions [41,42], which require high quality calibration and validation data that unfortunately are scant for remote regions with no commercial interest such as the treed peatlands of northern Canada.

Peatlands make up around 24% of boreal forests worldwide [43,44], and are prominent in Canada where they cover 13% of the country's land surface [45,46]. Black spruce (*Picea mariana* L.) is a dominant species in peatlands [43,47,48]. Even though roughly 70% of black spruce biomass is above ground [47], AGB makes up only a small portion of the total carbon stored in black spruce peatlands (~3% of the total soil carbon according to data from Bona et al. [49] and using the mean estimates from the model given in Bona et al. [50]). The main reason is that black spruce growing in peatlands are often stunted due to waterlogging during most of the growing season. Even if they are small trees, the large extent of these ecosystems (over 100 million hectares of Canada's boreal forest [51]) makes peatland black spruce AGB a significant carbon sink. Furthermore, of all the carbon pools in Canada's boreal forest, AGB is the most spatially variable, and the one that fluctuates the most because of its vulnerability to wildfires and other disturbances [52]. For example, an experimental fire in an Albertan black spruce peatland resulted in 100% tree mortality with around 25% of biomass being combusted and the remainder being added to the dead wood carbon pool, whereas only ~1% of the peat was lost [53]. This variability highlights the need for methods that can provide accurate AGB estimates for black spruce in peatlands.

In this study, we developed allometric equations specific to individual black spruce trees shorter than 5 m tall in peatlands of the Taiga Plains Ecozone. We used various model forms and TLS-measured tree attributes, and assessed which combinations led to the best estimates of AGB. Of particular interest to us were the models that rely on predictors that can also be measured by ALS (assuming high enough point density). We also assessed the use of quantitative structure models (QSMs) for measuring tree attributes such as DBH, height, and volume of small black spruce trees as this has been successfully performed for mature deciduous trees [20,54,55]. Finally, we compared the AGB estimates made by our best models with those made using the equations given by Ung et al. [15] and with those given in Bhatti et al. (another set of equations based on unpublished data specifically for black spruce less than 3 m tall) [56].

## 2. Materials and Methods

### 2.1. Study Area

This study was conducted in boreal forest peatlands located in the Northwest Territories (NWT) in and between Hay River, NWT and Fort Simpson, NWT (Figure 1). The study area lies within the mid-boreal Taiga Plains ecoregion [57], which has average annual temperatures between 1 and 4.5 °C, and a mean annual precipitation between 400 and 460 mm (mostly summer precipitation) [58]. Peatlands in this ecoregion are in the form of flat-topped, peat-rich areas elevated from the surroundings by underlying ice-rich permafrost (peat plateaus), smaller mounds of peat with permafrost and minerals in their cores (palsas), wetlands with parallel rows of peat material (northern ribbed fens), and wetlands with uniformly spread peat material (horizontal fens) [58].

We selected 10 circular plots that represented a variety of tree heights and densities typical of black spruce peatlands in the study area (Figure 1). Plots were 7.98 m in diameter (50 m$^2$) and contained anywhere from 23 to 115 trees per plot, most of which were black spruce, but with several (20 out of 606) tamarack (*Larix laricina* (Du Roi) K. Koch) as well.

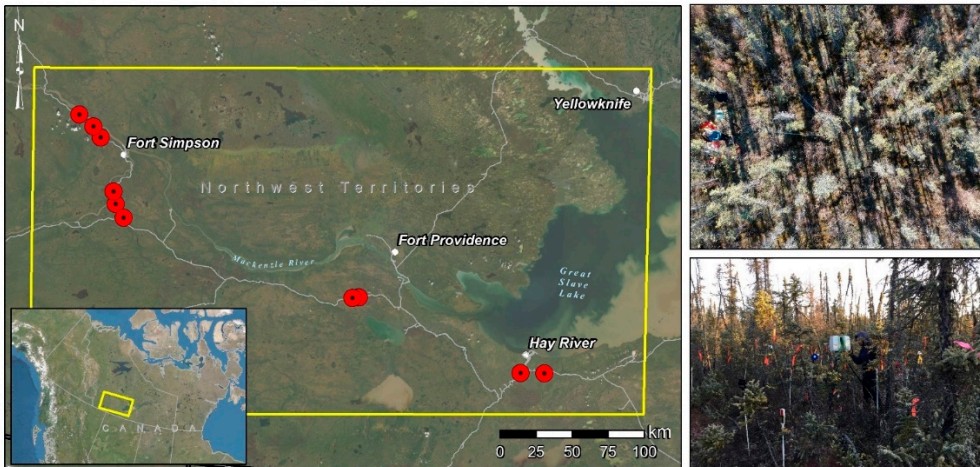

**Figure 1.** Field plots used in the study. Bottom left: Location of the study area within Canada. Centre: Location of the plots within the study area. Top right: A drone overhead view of one of the plots. Bottom right: Ground view of a plot, also Subfigure 2, 3. Scanning the Plots.

### 2.2. Plot Characteristics

All trees within the plot were flagged with orange flagging tape, and trees on the edge were flagged with pink. Their DBHs were measured using a diameter tape or electronic calliper for small trees, and their height with a Haglof Vertex IV and Transponder T3, or metallic measuring tape (for trees shorter than 2m). Ten black spruce trees representing the range of heights found in the plot were then selected for destructive sampling. To identify sample trees, reflective tape was wrapped around the trunk near 1.3 m, blue flagging tape was added to the branches, and a reflective marker stick with a number from 0 to 9 was placed next to the trunk. The trees selected for destructive sampling were also measured for their distance and bearing from the centre of the plot to help us find them in the resultant point clouds later.

All plots were scanned using a Leica C10 Terrestrial Laser Scanner from five different stations: one at the centre of the plot and four at points corresponding to the corners of a square that encompassed the plot. This follows the findings of Abegg et al. that the plot centre provides the best visibility in a plot, and additional scan locations placed evenly around the plot will reduce occlusion [26]. The C10 can produce colourized point clouds, for which the C10 camera was set to medium image resolution (960 × 960 px). The scan rate for the C10 laser instrument is 50,000 pts/sec, and it has a footprint diameter of 4.5 mm at 50 m range. Scan angles were set to 360° horizontally, and from −45° to 90° vertically. Five TLS targets were placed around the plot so they could be seen from each scan station. These targets were used to align the five scans during the process of registration, where all the scans were combined into a single point cloud of the entire plot. Finally, the plots were photographed by the C10 and by a 360° GoPro from each scan location, as well as from above using a drone to provide extra assistance in locating the destructively sampled trees in the point clouds later.

### 2.3. Destructive Sampling and Biomass Measurements

After each plot was scanned and photographed, the 10 trees selected for destructive sampling were cut down as close to the ground as possible. The trees were then cut into smaller pieces and put in bags marked with the tree's information (plot and tree number) for transportation. Some of the bags were brought to Edmonton, Alberta, and were left to dry in a storage area at 60 °C until there were no significant differences in weight measurements from day to day, with a minimum sitting time of at least one week. The remainder of the bags were weighed in Hay River, NWT and were dried at 65 °C for at least 1 week. The trees were then separated into main stem, branches, needles, and cones and weighed to the closest 100th of a gram. The individual weights of these components

will be used in a future study. The total weights of these components were added up to produce the reference values of AGB for each tree. The AGB distribution by both height and DBH of trees used in this study can be seen in Figure 2. Plot statistics for all predictors used in this study can be found in the Supplementary Materials (Table S1).

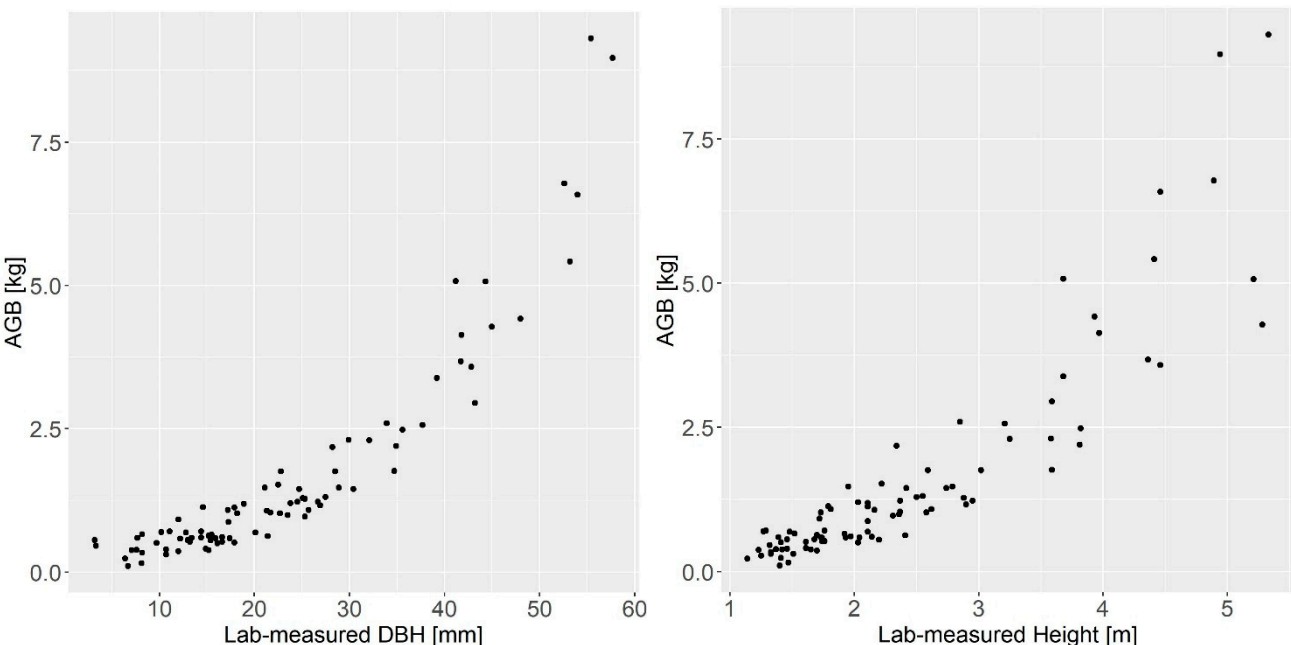

**Figure 2.** Scatterplots showing lab-measured diameter at breast height (DBH) versus above ground biomass (AGB) (**left**) and lab-measured height versus AGB (**right**) for the 89 trees used in this study.

### 2.4. Point Cloud Processing, Tree Extraction, and Height Measurements

Registration of the point clouds was performed in Leica's Cyclone software [59]. Each scan was imported into the software and the targets were used as anchor points for combining the individual scans. We required the difference in target location to be less than 6 mm when combining two scans, otherwise the constraint for those two targets was disabled for the purposes of registration. Once the scans were registered, the resulting point cloud was imported into CloudCompare [60] (an open-source point cloud editing software) for further analysis.

In CloudCompare, each plot point cloud was thinned using the cloud subsampling tool [60], with the minimum distance between points set to 1 cm to remove redundant points and reduce file size to about 15% of the original size, which helped increase computing speed for the next steps. The point cloud was cropped at the plot's circumference using the interactive segmentation tool to clip an area with a radius of 3.99 m from the plot centre [60]. To eliminate any points created by atmospheric debris or false returns, a statistical outlier filter [60] was applied, computed as:

$$T = \mu_d + nsigma \cdot \sigma_d \tag{1}$$

where $T$ is the threshold for removal, $\mu_d$ is the mean average distance from each point to its 10 nearest neighbours, *nsigma* is the standard deviation multiplier, and $\sigma_d$ is the standard deviation of the average distances of all points in the plot point cloud. This routine requires only an *nsigma* value to run and we used *nsigma* = 1.00. Points with an average distance to their 10 nearest neighbours exceeding the threshold ($T$) were removed (roughly 15% of the total points). The subsampled and filtered point clouds had a mean density of 33,000 points/m$^2$.

The trees that were selected for destructive sampling were then visually identified in the point cloud using the numbered reflective marker sticks and the distance and bearing

measurements taken in the field as guides. When these clues were not sufficient to identify the tree, the GoPro images were consulted to search for the differently coloured flagging tape on selected trees. These trees were manually clipped from the plot point cloud using the interactive segmentation tool in Cloud Compare [60], and the resulting clipped area was cleaned using the same tool so that only points from the selected tree remained. The cleaned clouds were then saved as individual tree point clouds for further analysis. The tree point clouds were manually straightened when necessary (i.e., the stem was aligned to the z axis when the tree was leaning) and, following Calders et al. [54], tree height was measured as the distance between the maximum and minimum z-coordinates of all the points in each individual tree point cloud.

### 2.5. Crown Diameter and Crown Area Measurements

Initially, we estimated crown diameter as the mean of two orthogonal pseudo-diameters (Supplementary Materials, Appendix S1), but because the crowns of these small trees are irregular, we decided to use a method that relied on 2D rasters to measure crown area and from that, derive crown diameter. We chose 1 cm as the size for the raster cells because it is the minimum distance between points after the point cloud thinning outlined in Section 2.4. We then created a count raster where the digital number (DN) in each cell was the number of points inside the square vertical prism represented by the cell. An initial estimate of the crown area was then obtained as the sum of the areas of non-empty cells in that raster. To reduce crown area overestimation caused by cells along the crown perimeter with very few or just one point, non-empty cells were then sorted in ascending order by DN, and the first 1% of cells in the ordered list were set to DN = 0 in the count raster. While this had little to no effect at the 1 cm cell level, it became more important when we analyzed the effect of raster cell size in the experiment outlined in Section 2.9. For the purposes of precisely calculating crown area and estimating the uncertainty of the crown area measurements, we determined which cells were on the perimeter and which were inside the crown by looking at each cell's 4-neighbours (neighbouring cells above, below, and on either side of the cell, but not diagonally adjacent). If the DN $\neq$ 0 in all four neighbouring cells, the cell was classified as an inner cell, otherwise the cell was classified as a perimeter cell. Then crown area was estimated as follows:

$$CA = (p \cdot 0.5a) + (I \cdot a), \tag{2}$$

where $p$ is the number of perimeter cells, $I$ is the number of inner cells, and $a$ is the area of a single cell.

The uncertainty of this measurement comes from not knowing whether the points in a perimeter cell are evenly distributed horizontally or if they are situated only on the side of the cell closest to the rest of the crown. Assuming the points were located at the centre of the cell, we can account for this uncertainty as follows:

$$\delta CA = \pm(p \cdot 0.5a). \tag{3}$$

Finally, we calculated crown diameter as the diameter of a circle of area equal to CA:

$$CD = 2\sqrt{CA/\pi}, \tag{4}$$

and derived the uncertainty of this measurement as:

$$\delta CD = (\delta CA \cdot CD)/(2 \cdot CA \cdot \pi). \tag{5}$$

(N.B. Uncertainty propagation for predictor measurement uncertainties and model parameters is explained in Appendix A).

### 2.6. TreeQSM Estimates of Height, DBH, and Volume

To evaluate the ability of QSMs to derive estimates of allometric variables, we used TreeQSM [61], an open-source MATLAB package that can be used to build QSMs from point cloud data. TreeQSM provides estimates for height, DBH (as the diameter of the cylinder fitted to the point cloud from 1.1 to 1.5 m), and volume. Each tree point cloud was run through the script five times to mitigate the random nature of the resulting QSMs created by the program, following the practices of other studies that generated multiple QSMs of individual trees to obtain more accurate estimates of tree attributes [62,63]. The attributes and their uncertainties were calculated using the average from the five QSMs for each tree. These estimates were then used as predictors in some of the AGB models we tested. Information on the input parameters used for this step can be found in the Supplementary Materials (Table S2).

### 2.7. Bounding Box Volume

Bounding box volume was measured as the volume of the smallest box that encompassed the entire tree point cloud. This was undertaken to expand on the work by Flade et al. who developed methods to use bounding box volume as a predictor for peatland shrub AGB [64]. The dimensions of the bounding box can be found by calculating the difference between the maximum and minimum coordinates on each axis. This is performed automatically in CloudCompare [60], so we used the box dimensions reported for the individual tree point cloud.

### 2.8. Fitting and Testing the Models

Our measurements gave us values for several different variables that could be used as predictors for our AGB models. The nine variables (or products of variables) used in our models were crown area (CA), crown diameter (CD), height as measured by the TLS (H), the product of crown area and height (CAxH), the product of crown diameter and height (CDxH), DBH as measured by the QSM (DBH), the product of DBH and height (DBHxH), QSM-measured volume (V (QSM)), and bounding box volume (V (Bounding Box)). Measurement uncertainties were recorded for each of these variables and combined with model uncertainties using the linear approximation and error propagation formulas outlined in Appendix A. These uncertainties were then applied to the final estimates of AGB as error bars (Supplementary Materials, Figures S1–S4). Confidence intervals for the models were measured at the 95% confidence level. Correlation coefficients were also calculated between all individual predictors and lab-measured AGB.

All models were fitted using R's lm() function [65] with equations representing three different types of models (Table 1). The input formula for power models was $log(y) \sim log(x)$ and the resulting coefficients were algebraically converted to fit the power function seen in Table 1. Similarly, multiple regression power models used the input formula $log(y) \sim log(x_1) + log(x_2)$. Quadratic models also required a log transformation of both $x$ and $y$ values before being fit into the lm() function using the formula $y \sim x + x^2$. Estimates were made using the resulting coefficients and then back transformed to give an estimate of AGB. To account for bias in the back-transformed models, the estimates were multiplied by a correction factor of $\varepsilon = e^{MSE/2}$, where *MSE* is the mean squared error of the fitted models with log-transformed variables [64,66,67].

**Table 1.** Forms of the equations used in the different models of this study. Tree attributes are denoted as $x$, and constants are denoted as $\alpha$, $\omega$, and $\beta$ in quadratic models. Constants are denoted as $\beta$ and exponents are denoted as $\alpha$ and $\omega$ in power equations.

| Model Type | Equation |
|---|---|
| Quadratic | $y = exp(\alpha x^2 + \omega x + \beta) \cdot \varepsilon$ |
| Power | $y = \beta \cdot x^{\alpha} \cdot \varepsilon$ |
| Multiple Regression Power | $y = \beta \cdot x_1{}^{\alpha} \cdot x_2{}^{\omega} \cdot \varepsilon$ |

Models were fitted using both ordinary and weighted least squares methods (OLS and WLS, respectively). OLS models were tested for heteroskedasticity using the Breusch–Pagan test [68,69]. For the weighted methods, the weight of each tree $i$ was inversely proportional to the number of trees $n_i$ with dry biomass within 1 kg of that tree:

$$W_i = (N - n_i) / \sum (N - ni), \tag{6}$$

where $N$ is the total number of trees in the sample. This gives more weight to bigger trees in the sample, which are less represented than smaller trees. Weights also helped reduce the heteroskedasticity observed in the residuals of some of the OLS models.

We assessed which of these fitted models performed the best using a 10-fold cross validation [70] with each plot acting as a fold to create a scenario analogous to using the model in a non-sampled location and therefore assess its transferability [71]. This method used the trees from 9 out of 10 plots to fit the model and then tested it on the trees of the left-out plot. We then fit the model again using a different combination of nine fitting plots and tested it on a different plot than the previous iteration(s). The process was repeated until each plot was used for testing once. During each iteration, we recorded the mean average error (MAE), adjusted $R^2$ (adj. $R^2$), and root mean square error (RMSE).

We then used the average of each of these metrics from the completed cross validation to rank the models from 1 to 42. We also noted the coefficient of variation in RMSE to provide insight on the robustness of the model when it is exposed to new data from different areas of peatland. We then selected the best model for each of the nine predictors above for further analysis.

To assess how our top model (best of the best models) fares compared with the published equations of Ung et al. [15] and Bhatti et al. [56] for commercial and small black spruce, respectively, the individual AGB of each tree harvested in this study between 1.3 and 3 m tall (as the equations of Bhatti et al. are only applicable to trees shorter than 3 m) was estimated using lab measurements of height and DBH and those equations. There are also AGB equations for black spruce in the Northwest Territories specifically, but they are only suited to trees with DBH > 6 cm [72] and therefore were not used in this study. The estimates from the published equations were compared with lab-measured AGB and the resulting RMSE and coefficient of determination were calculated and compared with those from the leave-one-plot-out cross validation of our top model. That is, to allow for a fair comparison, our predicted value for each of the trees ≤ 3 m tall came from the version of our top model that was fitted using all plots except the one from which the tree was harvested.

### 2.9. Surrogate Point Density Sensitivity Analysis

To assess how decreasing point density would reduce the accuracy of the AGB estimates in a scenario where the point clouds used for tree measurements come from ALS instead of TLS, an exploratory sensitivity analysis was performed using the cell size of the rasters as a proxy for point density of first returns. In addition to the 1 cm cell rasters, we created rasters with cells of increasing size at 5 cm steps up to 50 cm. Assuming one point per cell, this would act as a surrogate for point densities ranging from 10,000 to 4 pt/m$^2$. Crown area and crown diameter were estimated for each raster using the methods outlined in Section 2.6 for the 1 cm cell rasters. Recomputing tree height for decreasing point densities is not as straightforward and would require real data from ALS or DLS (drone ALS) [73], so we decided to forego the assessment of the impact on tree height.

We then calculated the crown area of each tree for each raster cell size and used the equations provided by the best model for crown area and height to estimate AGB; we compared these results with the lab-measured AGB. We recorded the RMSE and the coefficient of determination to see how these metrics would behave with decreasing point density.

## 3. Results

In total, we fitted 42 models to our data using different combinations of TLS-measured predictor variables, model forms, and model fitting methods to estimate AGB. Of the 100

trees harvested in this study (10 in each of the 10 plots), 89 were used in the fitting and final analysis of our models (Table 2). Those 89 trees had an average AGB of 1.63 kg with a range of 0.11 to 9.31 kg. Details of the trees that were excluded from modelling and the reasons for their exclusion can be found in the Supplementary Materials (Table S3). Occlusion was the most common issue that led to trees being excluded from the analysis. Figure 3 shows examples of occluded and unoccluded point clouds.

**Table 2.** Field-measured height and DBH of all trees within a plot as measured in the field as well as for the sample (harvested) trees used in this study. Shown as average height ± standard deviation (minimum value in range; maximum value in range). AGB = above ground biomass, DBH = diameter at breast height.

| Plot Name | # * of Black Spruce in Plot | Height for All Trees (m) | DBH for All Trees (cm) | TLS Measured Crown Area (m²) | Height for Sample Trees (m) | DBH for Sample Trees (cm) | Avg AGB for Sample Trees (kg) |
|---|---|---|---|---|---|---|---|
| V2B006 | 52 | 2.6 ± 1.0 (1.3; 5.7) | 2.9 ± 1.5 (0.5; 7.1) | 0.16 ± 0.11 (0.05; 0.43) | 2.6 ± 1.0 (1.6; 5.0) | 2.8 ± 1.3 (1.5; 6.0) | 1.55 ± 1.50 (0.41; 5.42) |
| V2B009 | 47 | 2.4 ± 1.0 (1.3; 5.4) | 2.4 ± 1.4 (0.3; 6.2) | 0.18 ± 0.09 (0.06; 0.35) | 2.7 ± 1.1 (1.4; 5.1) | 2.9 ± 1.5 (1.1; 6.2) | 1.93 ± 1.95 (0.37; 6.78) |
| V2B011 | 31 | 2.7 ± 1.3 (1.3; 6.2) | 2.9 ± 1.8 (0.3; 6.5) | 0.23 ± 0.10 (0.08; 0.41) | 2.7 ± 1.2 (1.3; 4.7) | 2.9 ± 1.6 (0.3; 5.1) | 1.85 ± 1.19 (0.46; 4.14) |
| V2B012 | 23 | 3.4 ± 1.7 (1.4; 7.7) | 3.7 ± 2.4 (0.6; 9.7) | 0.32 ± 0.21 (0.11; 0.66) | 3.4 ± 1.5 (1.5; 5.6) | 3.7 ± 2.1 (0.9; 6.7) | 3.80 ± 3.40 (0.60; 9.31) |
| V2B015 | 44 | 2.0 ± 0.7 (1.4; 4.6) | 2.0 ± 1.0 (0.3; 5.9) | 0.15 ± 0.10 (0.04; 0.35) | 2.2 ± 0.9 (1.6; 4.4) | 2.3 ± 1.2 (0.3; 4.7) | 1.02 ± 1.06 (0.11; 3.68) |
| V2B016 | 32 | 3.0 ± 1.2 (1.3; 5.8) | 3.1 ± 1.7 (0.4; 6.6) | 0.17 ± 0.05 (0.08; 0.23) | 2.9 ± 1.4 (1.3; 5.5) | 2.8 ± 1.6 (0.5; 5.4) | 1.72 ± 1.52 (0.28; 5.07) |
| V2B019 | 92 | 2.3 ± 0.7 (1.3; 5.0) | 1.9 ± 1.1 (0.3; 5.7) | 0.11 ± 0.05 (0.06; 0.22) | 2.4 ± 0.8 (1.4; 3.8) | 2.1 ± 1.0 (0.7; 4.1) | 0.93 ± 0.69 (0.24; 2.48) |
| V2B022 | 25 | 2.3 ± 0.7 (1.5; 4.7) | 2.3 ± 1.3 (0.6; 6.3) | 0.18 ± 0.11 (0.08; 0.49) | 2.4 ± 0.9 (1.5; 4.7) | 2.4 ± 1.5 (1.0; 6.3) | 1.50 ± 1.73 (0.51; 6.58) |
| V2B023 | 112 | 2.2 ± 0.8 (1.3; 7.1) | 2.1 ± 1.2 (0.3; 6.9) | 0.10 ± 0.07 (0.04; 0.27) | 2.5 ± 1.1 (1.5; 4.8) | 2.3 ± 1.4 (0.7; 4.8) | 1.10 ± 1.24 (0.15; 4.28) |
| V2B026 | 115 | 2.0 ± 0.6 (1.3; 4.6) | 1.8 ± 1.1 (0.3; 5.2) | 0.12 ± 0.05 (0.06; 0.20) | 2.2 ± 0.6 (1.6; 3.7) | 2.3 ± 1.2 (1.2; 5.2) | 0.95 ± 0.81 (0.34; 2.95) |
| Total | 573 | 2.3 ± 1.0 (1.3; 7.7) | 2.3 ± 1.4 (0.3; 9.7) | 0.17 ± 0.12 (0.04; 0.66) | 2.6 ± 1.1 (1.3; 5.6) | 2.6 ± 1.5 (0.3; 6.7) | 1.63 ± 1.83 (0.11; 9.31) |

* # = number.

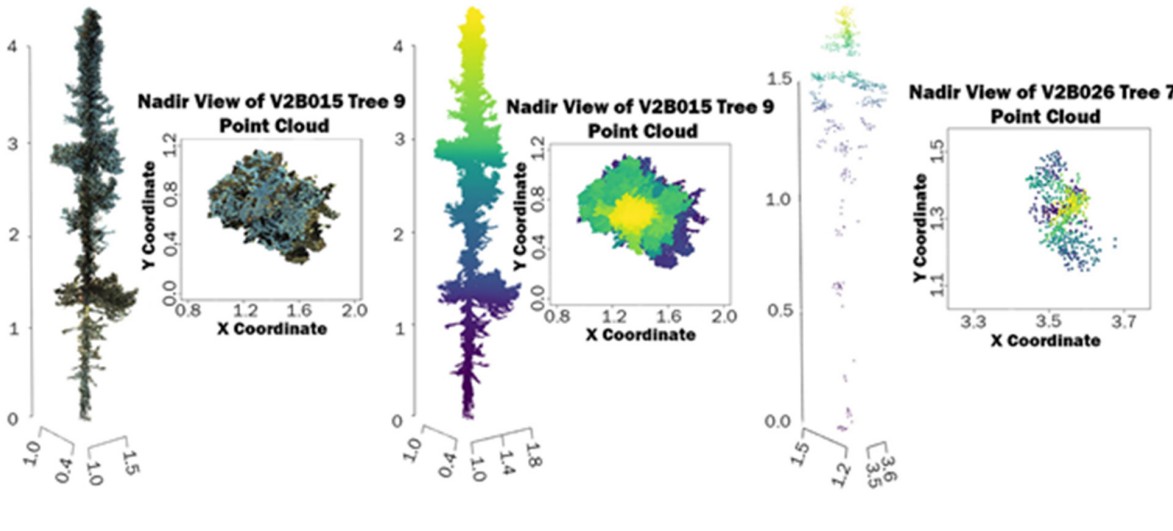

(**A**)       (**B**)       (**C**)

**Figure 3.** A comparison of an unoccluded tree point cloud (left) versus an occluded one (right). (**A**) A true colour view (silhouette and overhead) of the point cloud corresponding to harvested tree #9 from plot V2B015; this point cloud is full, detailed, and shows the complete tree structure, making it easy to obtain measurements from it. (**B**) Same as (**A**) but using a colour ramp based on height. (**C**) Harvested tree #7 from plot V2B026; this point cloud is incomplete because of occlusion, making it difficult to use in our workflow.

### 3.1. Effect of Weights on Final Models

In our sample, 73 trees had a lab-measured AGB of less than 2.50 kg (Figure 2), meaning larger trees were underrepresented when fitting the models. Giving more weight to larger trees in the WLS estimation of model parameters saw our model fits either improve or remain constant (adjusted $R^2$ and RMSE) across the board (see Figure 4 for one example). In this model (the bounding box volume power model), the *p* value of the Breusch–Pagan test for the residuals in the OLS method was 0.14, indicating that the residuals are homoscedastic. Even so, adding weights to give each interval of the AGB range equal influence on the model resulted in an improved adjusted $R^2$ (0.89 for WLS, 0.86 for OLS), and a constant RMSE (0.66 kg for both).

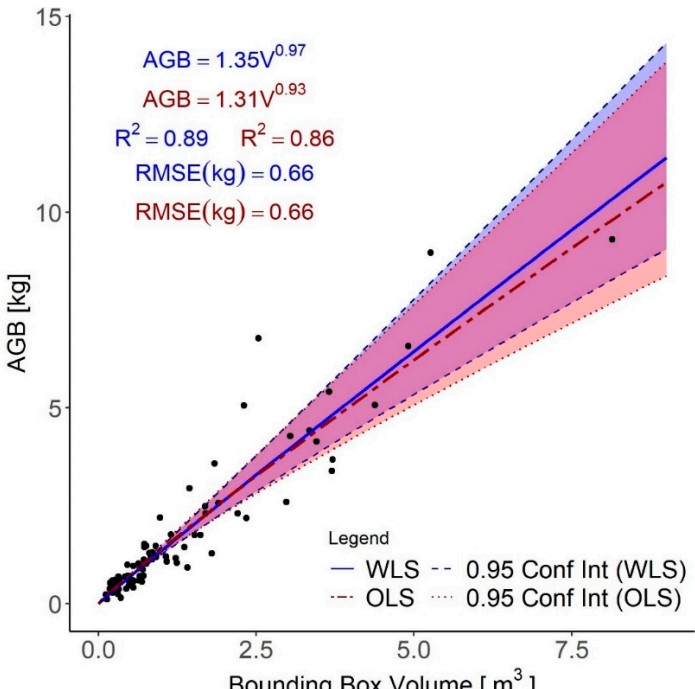

**Figure 4.** Comparison of models fitted using the weighted least squares (WLS) method (blue solid line) and the ordinary least squares (OLS) method (red dash-dotted line). Both models use the bounding box volume of individual tree point clouds as a predictor, but the WLS model gives more weight to the trees with higher above ground biomass. The dashed and dotted lines show the 0.95 confidence intervals for the WLS and OLS models, respectively.

### 3.2. QSM Effectiveness

Each of the 89 trees in our study was also run through the TreeQSM script [61] as outlined in Section 2.7. The average measurement uncertainty for total tree volume was 1.27 L, roughly 9% of the average total volume. Similar relative uncertainties were observed for DBH (8%) and stem volume (10%). DBH estimation proved unreliable when compared with lab-measured results. The coefficient of determination for observed (lab measured) versus predicted (QSM) DBH was 0.58, with an RMSE of 2.7 cm, and the bias was +2.0 cm (relative to the mean observed DBH, this is 115% and 86%, respectively). Height estimation from the QSMs was more reliable than for DBH, and close to the simple max($z$)–min($z$) calculation. QSM-obtained heights returned a coefficient of determination of 0.97 and an RMSE of 0.20 m, while the max–min method returned a coefficient of determination of 0.97 and an RMSE of 0.18 m. In terms of AGB, estimates given by our best model (crown area and height multiple regression power model, outlined in Section 3.3) gave an average adjusted $R^2$ value of 0.94 and an average RMSE of 0.34 kg (21% of the average AGB of the sample), whereas the best model using only QSM-measured predictors (Volume WLS

power model) gave an average adjusted $R^2$ of 0.82 and an average RMSE of 0.70 kg (43% of the average AGB of the sample).

### 3.3. Model Rankings

The multiple regression power models that used the product of crown size (crown area or crown diameter) and height yielded the best results (Table 3). Because crown diameter was a parameter derived from crown area, these two power models returned the same results. A graphical representation of the fitted model using the product of crown area and height can be seen in Figure 5. The fact that the multiple regression models use a separate exponent for each factor makes for a closer fit than in the normal power models where a single exponent applies to the completed product of the factors.

**Table 3.** Final rankings of the best models built for each set of predictors outlined in Section 2.9. Rankings were based on average mean absolute error (MAE), average root mean squared error (RMSE), and average adjusted $R^2$ obtained from the leave-one-plot-out cross validation outlined in Section 2.9 (the full rankings of all 42 models can be seen in Table S7 of the Supplementary Materials). The numbers in brackets correspond to the value of the metric when all plots (89 trees in total) are used to fit the model. Multi Pwr = multiple regression power, Pwr = power, Quad = quadratic, CA = crown area, CD = crown diameter, H = height, V (Bounding Box) = bounding box volume, DBH = diameter at breast height, V (QSM) = QSM-derived volume, QSM = quantitative structure model.

| Model Type | Model Predictors * | Avg MAE | Avg RMSE (kg) | Avg Adj $R^2$ | Final Ranking |
|---|---|---|---|---|---|
| Multi Pwr | CAxH and CDxH | 0.22 (0.21) | 0.34 (0.36) | 0.94 (0.94) | 1 |
| Pwr | V (Bounding Box) | 0.40 (0.39) | 0.59 (0.66) | 0.89 (0.89) | 2 |
| Pwr | H | 0.45 (0.41) | 0.63 (0.67) | 0.88 (0.88) | 3 |
| Multi Pwr | DBHxH | 0.46 (0.41) | 0.64 (0.67) | 0.88 (0.88) | 4 |
| Pwr | V (QSM) | 0.50 (0.46) | 0.70 (0.85) | 0.82 (0.82) | 5 |
| Pwr | CA and CD | 0.67 (0.66) | 0.95 (1.04) | 0.71 (0.71) | 6 |
| Quad | DBH | 0.83 (0.75) | 1.18 (1.30) | 0.66 (0.66) | 7 |

* All models in this table were fitted using weighted least squares.

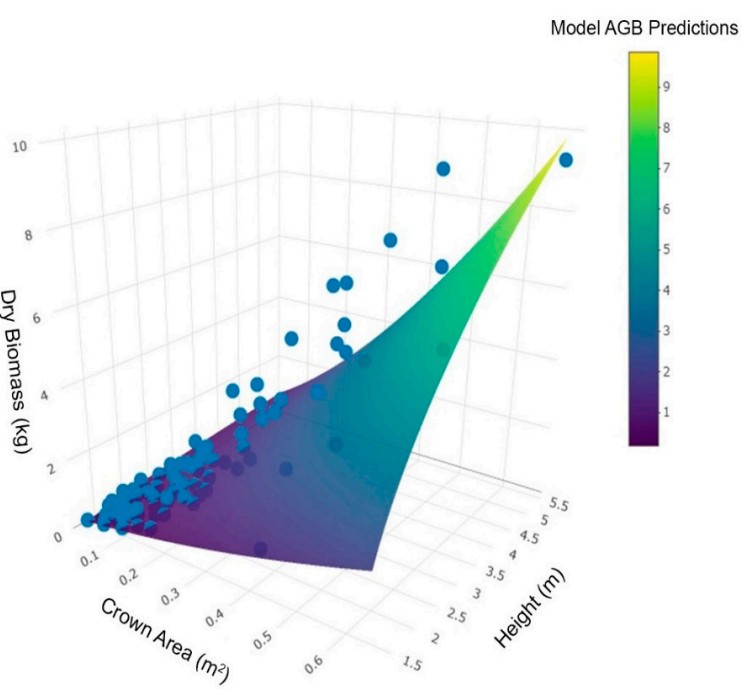

**Figure 5.** Graphical representation of the best model (multiple regression crown area and height weighted least squares power model) results. The curved surface represents the model predictions for every value of crown area and height, with colour as an additional indicator of biomass value. The blue dots represent the observed values for the trees used in this study. AGB = above ground biomass.

Model parameters and standard errors are reported for each type of model in Table 4 (for full results refer to the Tables S4–S6 in the Supplementary Materials). The standard errors were combined with the recorded measurement uncertainties and plotted in actual AGB versus estimated AGB plots, which can be found in the Supplementary Materials (Figures S1–S4). Methods and equations for estimating the uncertainty of estimates appear in Appendix A.

**Table 4.** Model coefficients and their standard errors for models in Table 3. AGB = above ground biomass, Multi Pwr = multiple regression power model (AGB = $\beta \cdot x_1{}^\alpha \cdot x_2{}^\omega \cdot \varepsilon$), Pwr = power model (AGB = $\beta \cdot x^\alpha \cdot \varepsilon$), Quad = quadratic model (AGB = $exp(\alpha x^2 + \omega x + \beta) \cdot \varepsilon$), CA = crown area, H = height, CD = crown diameter, DBH = diameter at breast height, V (Bounding Box) = bounding box volume, V (QSM) = QSM-derived volume.

| Model Type | $x_1$ | $x_2$ | $\beta$ | $\beta$ std err | $\alpha$ | $\alpha$ std err | $\omega$ | $\omega$ std err |
|---|---|---|---|---|---|---|---|---|
| Multi Pwr | CA | H | 0.73 | 0.13 | 0.54 | 0.06 | 1.68 | 0.09 |
| Multi Pwr | CD | H | 0.64 | 0.10 | 1.07 | 0.11 | 1.68 | 0.09 |
| Multi Pwr | DBH (QSM) | H | 0.16 | 0.02 | 0.06 | 0.07 | 2.19 | 0.14 |
| Pwr | V (Bounding Box) | - | 1.35 | 0.05 | 0.97 | 0.04 | - | - |
| Pwr | H | - | 0.16 | 0.02 | 2.29 | 0.09 | - | - |
| Pwr | V (QSM) | - | 0.23 | 0.03 | 0.76 | 0.04 | - | - |
| Pwr | CA | - | 14.64 | 2.40 | 1.29 | 0.09 | - | - |
| Pwr | CD | - | 10.73 | 1.55 | 2.57 | 0.18 | - | - |
| Quad | DBH (QSM) | | −0.97 | 0.13 | 0.40 | 0.08 | 0.25 | 0.15 |

All models in this table had $\varepsilon$ = 1.00.

### 3.4. Comparisons with Published AGB Equations for Black Spruce

We used a subset of 64 trees to compare our top model's estimates with those made using equations from both Ung et al. [15] and Bhatti et al. [56]. We could not use the whole sample for this comparison because the equations in Bhatti et al. are recommended for trees less than 3 m in height [56] and because four trees from our sample had no lab-measured DBH because their reconstructed lab-measured heights were less than 1.3 m. Our best model (crown area and height WLS multiple regression power model) outperformed both the other models, even though it does not use DBH and despite the fact the RMSE for our model was based on the leave-one-plot-out cross validation. Our estimates had the lowest RMSE of the tested methods at 0.21 kg or 24% of the average biomass of this subset of trees, compared with 0.31 kg (36%) for the Ung et al. estimates, and 0.35 kg (40%) for the Bhatti et al. estimates (Figure 6). The crown area and height model also performed better than another multiple regression WLS model fitted using lab-measured DBH and height (not shown).

We also compared the results from the cross validation of the crown area and height multiple regression power model with the estimates given by the equations of Ung et al. [15] for the subset of all trees with lab-measured height and DBH (85 of the 89 in the sample). The average RMSE of our top model in this test was 0.39 kg (23% of the average AGB of this subset) compared with 0.51 kg (30%) in the estimates made using Ung et al.'s equations [15]. The $R^2$ was also higher in this test for our top model: 0.96 compared with 0.93 for Ung et al. estimates. When all plots (a total of 89 trees) were used for fitting the crown area and height model, the adjusted $R^2$ of the model was 0.94, and the RMSE was 0.36 kg (22% of the average AGB).

Of the predictors used in this study, bounding box volume ($r$ = 0.93), lab-measured DBH ($r$ = 0.90), and TLS-derived height ($r$ = 0.90) were most strongly correlated with AGB, while crown area ($r$ = 0.81) and QSM-measured DBH ($r$ = 0.67) showed the weakest correlation with AGB (Figure 7).

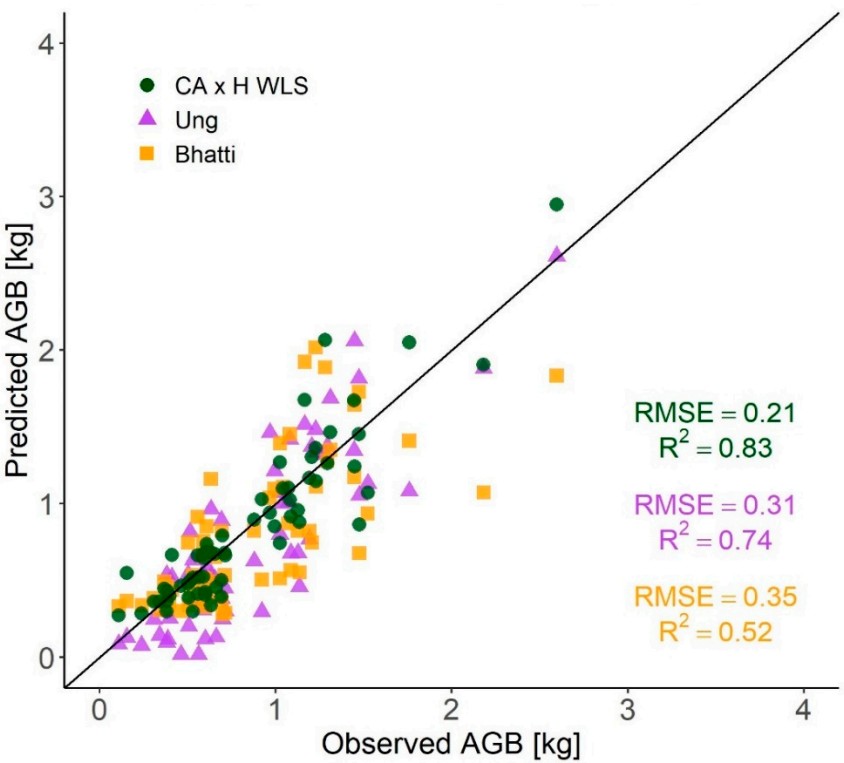

**Figure 6.** Comparison with lab-measured AGB of estimates made using our best terrestrial laser scanning (TLS) model (crown area and height weighted least squares multiple regression power model, green circles), Ung et al. [15] equations for black spruce (purple triangles), and the equations in Bhatti et al. [56] for small black spruce (orange squares) for the 64 trees between 1.3 and 3 m tall in our study. Each model's root mean squared error (RMSE) and coefficient of determination ($R^2$) are also shown (top: our model, middle: Ung et al., bottom: Bhatti et al.). Estimates from our model came from the leave-one-plot-out cross validation models. The black line denotes the 1:1 line.

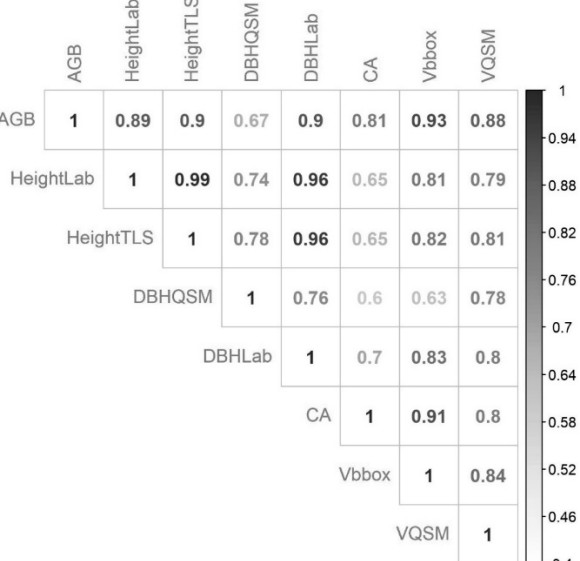

**Figure 7.** Correlation matrix between individual tree above ground biomass (AGB) and each predictor (lab-measured, terrestrial laser scanner (TLS) measured, or quantitative structure model (QSM)-derived), and between each pair of predictors. Stronger correlations are denoted with darker tone. Crown diameter is not included because it is equivalent to crown area. DBH = diameter at breast height, CA = crown area, Vbbox = bounding box volume, VQSM = QSM-derived volume.

### 3.5. Crown Area Sensitivity Analysis

The simulation of decreasing point density revealed that the AGB estimation errors in the crown area and height multiple regression power model remain low at densities above 16 pts/m$^2$ (Figure 8). While both RMSE and $R^2$ show some worsening over the full range of nominal point densities tested, most of this occurs for lower nominal point densities. RMSE of the AGB estimates increases from 0.36 kg when using 10,000 pts/m$^2$ to 1.26 kg when using 4 pts/m$^2$, but only 17% of this increase is seen between 10,000 and 16 pts/m$^2$. A similar pattern is also observed for $R^2$.

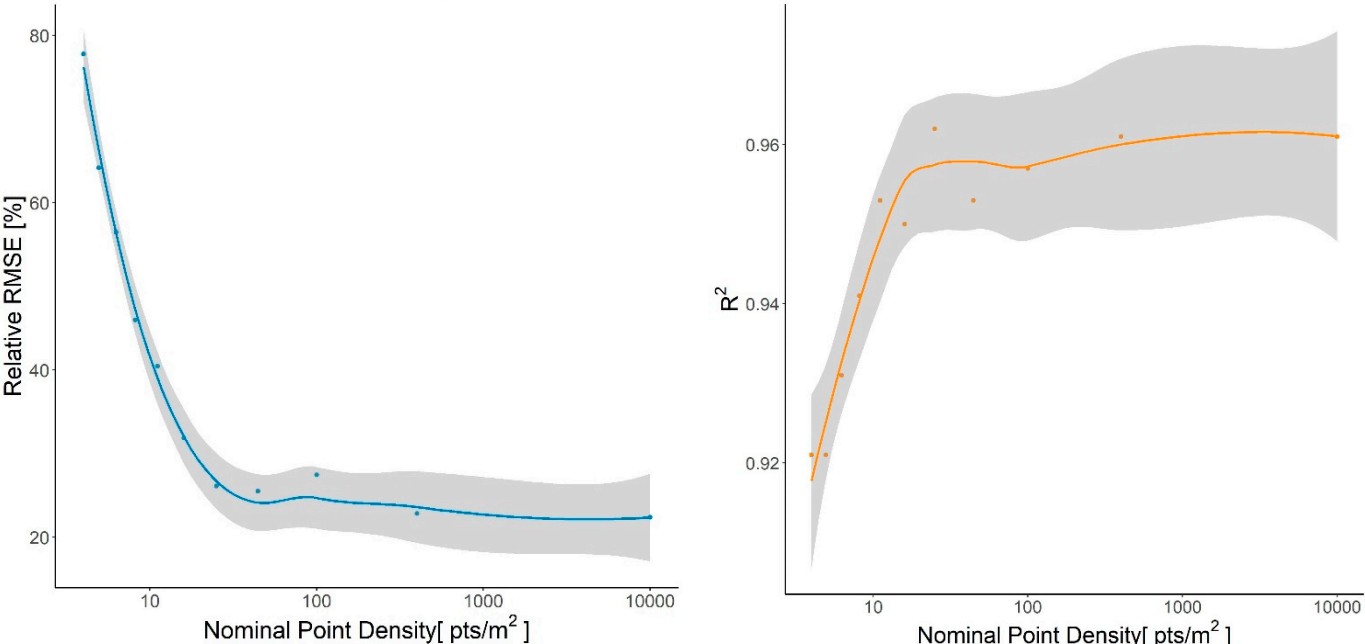

**Figure 8.** Relative root mean square error (**left**) and coefficient of determination ($R^2$) (**right**) values for above ground biomass predictions of the multiple regression power model made using the product of height and crown area derived from varying nominal point densities. Results are fitted with a loess (locally weighted smoothing) line to show trends. Grey areas show the 0.95 confidence interval for these lines.

## 4. Discussion

### 4.1. Effect of Weights on Final Models

The residuals of most OLS models did not show heteroskedasticity in a Breusch–Pagan test [68,69] (the only exception being the power model of the product of crown area and height). However, we did notice slightly larger residuals for trees in the higher end of our lab-measured AGB range when OLS was used instead of WLS. A possible reason for this is that there were more trees with low AGB (<2.50 kg) than there were trees with higher AGB (>2.50 kg). While it is to be expected that treed peatlands will have more smaller than larger trees [48], this situation does not translate well to model fitting, because if each point is given the same weight and more points are at the lower end of the range, then the model will tend to fit those points more, leading to larger residuals for the taller trees.

### 4.2. QSM Effectiveness

At the outset of the study, we expected that the QSMs could be used to reliably estimate AGB. Because the trees were small and frequently clustered together, there was often occlusion that caused anomalies in the final QSMs. We were also unable to remove the needles from the point clouds without removing many of the branches as well. The branches were often quite small and difficult to differentiate from the needles, particularly on the smallest trees. Stem reconstruction was partial and noisy for most of the trees, which meant that the DBH estimates were unreliable, with a relative RMSE of more than

100% of the average field-measured DBH, which propagated through to the QSM volume estimates. QSM volume was still used as a single predictor in one of the models, but it was outperformed by other TLS models that also used only single predictors, namely bounding box volume and height.

The QSMs did not provide the kind of consistent results other authors obtained for mature deciduous trees [54,63]. Occlusion played a large role in this, as the QSMs rely on being able to fit cylinders to segments of the point cloud [61], and when the stems of trees were occluded, it was difficult for TreeQSM to accurately fit cylinders to the stem and obtain DBH measurements. Part of the reason for this could have come from the plot-centered scan approach used in this study, which can cause occlusion on target trees from the understory and branches of other trees [74] and contrasts with the tree-centered approach used in most QSM studies. It should also be noted that QSMs produce their best results in leaf off conditions [75,76]. We were unable to remove points corresponding to needles effectively, as needles are often easily confused with woody structure [77,78]. Therefore, we had to run our QSMs with the needles still on in the point cloud, which probably also contributed to the poor results. The AGB estimation error for models that relied on QSM-measured predictors is not directly related to needles, but the thick branching and foliage created significant occlusion of the stem. In conclusion, we do not recommend the use of QSMs for AGB estimation of small black spruce trees that are close together as is often the case in boreal peatlands.

*4.3. Model Rankings*

The best TLS models were the multiple regression power models of crown area (or alternatively, crown diameter) and height, where the two predictors are multiplied, and each has a different exponent. The power models that assigned a single exponent to the product of multiple variables still performed well, but not as well as these models (Supplementary Materials, Table S7). The strength of the multiple regression models comes from their ability to be more flexible when fitting the curve to the data, leading to smaller residuals than the models using only a single exponent. We found that within the multiple regression power models, the crown area and height model performed better than the lab-measured DBH and height model. This finding was unexpected because DBH correlates more strongly with AGB than crown area. A possible explanation for this is in the correlation between predictors. In our sample, crown area and height are less correlated than DBH and height (Figure 4) ($r = 0.65$ to $r = 0.96$, respectively), which is probably the reason why a model using those two variables would be better able to explain the variance in AGB. These findings are consistent with other studies showing that crown size can improve the predictive capabilities of AGB allometric equations in other ecosystems [13,36,79].

Quadratic models performed slightly worse than power models in most situations except for the QSM-derived DBH. The latter was, however, the worst performing of all our best models (Table 3), probably because of the difficulty of obtaining accurate DBH measurements from the QSMs. When the same model was fitted using lab-measured DBH it had an RMSE of 0.43 kg, 68% lower than that of the QSM-derived DBH model (this comparison was undertaken using the subset of the 85 trees that had lab-measured DBHs). An important consideration for quadratic models is that, unlike power models, the regression line does not intercept the $y$ axis at $y = 0$. This means that it is entirely possible to predict negative AGB for small trees. It is possible to force the regression to have an intercept at $y = 0$, but because our intercepts were generally quite small, we found that doing this caused the model to yield lower adjusted $R^2$ and higher RMSE values for the trees that fell in the range of our sample, so we used the unconstrained models instead.

Since the vast majority of trees in our study were between 1.3 and 5 m tall, we recommend that our equations be used only for black spruce trees in that height interval. The equations published by Ung et al. [15] are likely to give better estimates for trees taller than 5 m.

The low coefficient of variation for RMSE in the leave one-plot-out-cross-validation (Supplementary Materials, Table S7) corroborated the good transferability potential of the models and helped alleviate concerns about overfitting. As such, we believe that they should work equally well in other boreal peatlands [71]; however, further study is needed to confirm this.

The model based on bounding box volume ranked third in the final rankings of TLS models (Table 3). Unlike in QSMs, where volume is calculated using an array of cylinders to represent the stem and branches of a tree, bounding box volume is easily computed from the maximum and minimum point coordinates in each axis. We selected this predictor following Flade et al. [64], who found that bounding box volume can predict the AGB of boreal shrubs with relative RMSE of 76% when compared with the average AGB of shrubs in their sample. We expanded on this idea by applying it to small black spruce trees and showed that bounding box volume can be a useful predictor for estimating tree biomass as well, with a relative average RMSE of 40% and an $R^2$ of 0.87. Bounding box volume could be useful in situations where the point cloud is not dense enough to reliably estimate crown area but where there is ancillary high-resolution imagery that can identify the individual small black spruce within the bounding box.

The model based on TLS-measured height alone was fourth in the ranking, outperforming the models that used QSM-derived attributes, as well as the models that used only crown size as a predictor. The fact that they performed better than the single-predictor models based on crown size indicates that height is the more important factor in the top two models, with crown size being a secondary piece of information that can improve on the models built using height alone.

### 4.4. Comparisons with Other AGB Estimation Methods

We found that our best models (using crown size and height as predictors) outperformed the other existing model for small black spruce (Bhatti et al.) [56]. The mean RMSE from the leave-one-plot-out cross validation of our crown area and height model for the subset of trees shorter than 3 m tall was 0.21, compared with 0.35 for the Bhatti et al. equations, even with the latter using lab-measured DBH and height. Furthermore, our best models also outperformed the DBH and height model from Ung et al. [15] for our sample and therefore could be a preferred alternative when DBH is not available or is hard to measure from TLS, such as where there are many small trees close together. When the AGB was estimated using Ung et al.'s equations with lab-measured height and DBH, the RMSE was 0.51 kg, 30% of the average AGB, still worse than our model: the average RMSE of our best model was 0.39 kg, 23% of the average AGB. Our best model requires that the point clouds are correctly segmented and classified, something that has been shown for tall trees [80–82], and with some success for small trees and shrubs when they are not part of an understory [83,84]. The accuracy of our models combined with their transferability potential to estimate AGB without setting foot on the ground (when using high-density ALS instead of TLS) makes our models an excellent tool with which to estimate the AGB of individual small black spruce in boreal forest peatlands.

### 4.5. Crown Area Sensitivity Analysis

The sensitivity analysis with increasing raster cell sizes as a proxy for decreasing point density suggests that our best models could perform similarly using ALS data instead of TLS, providing point density is greater than 16 pts/m$^2$. While the point density of ALS and DLS (laser scanning from drones) point clouds is dependent on the flight altitude and speed, number of flight lines, and scan and pulse rates, the 16 pts/m$^2$ threshold is attainable by both ALS and DLS [85–87]. This threshold is also consistent with the one suggested in a recent study for height estimation of coniferous trees using drone-based LiDAR point clouds of different point densities, which found that height accuracy only worsens at below 17 pts/m$^2$ [73]. Nevertheless, the observed accuracy loss in a real scenario could start at higher point densities because we assumed no effect on tree height and uniform horizontal

distribution of laser pulses. While this analysis provides some preliminary insight as to how the crown area component of our best model is affected by decreasing point densities, it is by no means extensive and further analysis using actual airborne data is needed. We are planning a future experiment with real ALS and DLS data to explore this issue.

## 5. Conclusions

Our best TLS models produced estimates of AGB for small black spruce that were more accurate than estimates derived from widely used allometric equations first published in Lambert et al. [20] and then updated in Ung et al. [15], which require time-consuming field measurements of DBH and height. Our top model uses as predictors tree height and crown size, but other models performed well also, with RMSE ranging from 21% to 73% and adjusted $R^2$ from 0.94 to 0.62. They have the advantage of not being reliant on DBH, which cannot be reliably measured from the air [6,36,38]. DBH can also be difficult to measure from the ground using TLS when the trees are small, compact, and close to each other. Instead, our models use predictors that have the potential to be measured from the air using high-density point clouds, from drone or airplane, photogrammetry or LiDAR. As such, the set of models presented here could provide a valuable tool for estimating the individual tree AGB of a small black spruce in boreal peatlands. In future studies we plan to scale up from the individual tree level to the plot level and assess the suitability of our models to estimate AGB density (Mg/ha) in black spruce peatlands using point clouds acquired from the air, including the effect of point density on AGB estimates both at the individual tree and plot levels.

**Supplementary Materials:** The following are available online at https://www.mdpi.com/article/10.3390/f12111521/s1, treedata.csv: CSV file with the individual tree data used to fit the models; Table S1: Average values for each predictor of the harvested trees used in this study; Appendix S1: Original Orthogonal Measurement Method for Estimating Crown Diameter; Table S2: TreeQSM input parameters used to build quantitative structure models (QSMs) of the individual trees; Figure S1: Scatterplots of predicted versus observed AGB for multiple regression models; Figure S2: Predicted versus observed AGB plots for power WLS models; Figure S3: Scatterplot of predicted versus observed AGB for power model using crown diameter as a predictor when fitted using the whole sample; Figure S4: Scatterplot of predicted versus observed AGB for quadratic WLS model using QSM-derived DBH as a predictor; Table S3: Trees removed from this study after being scanned and harvested, and the reason for discarding them; Table S4: Estimated value, 95% confidence intervals and standard errors of model parameters for multiple regression power models; Table S5: Estimated value, 95% confidence intervals and standard errors of model parameters for power models; Table S6: Estimated value, 95% confidence intervals and standard errors of model parameters for quadratic models; Table S7: Model rankings based on accuracy and goodness of fit statistics from the leave-one-plot-out cross validation.

**Author Contributions:** Conceptualization, G.C. and S.W.; methodology, S.W. and G.C.; software, S.W.; validation, S.W.; formal analysis, S.W.; investigation, S.W. and M.F.; resources, G.C. and G.A.S.-A.; data curation, S.W.; writing—original draft preparation, S.W. and G.C.; writing—review and editing, S.W., G.C., G.A.S.-A. and M.F.; visualization, S.W. and M.F.; supervision, G.C., G.A.S.-A. and M.F.; project administration, G.C.; funding acquisition, G.C. and G.A.S.-A. All authors have read and agreed to the published version of the manuscript.

**Funding:** This research was funded by the collaborative research agreement between the Canadian Forest Service and the Government of the Northwest Territories, CRA-R00893, and Sanchez-Azofeifa's National Science and Engineering Research Council of Canada (NSERC) Discovery Grant program.

**Data Availability Statement:** The data used to fit the models in this study are available in the treedata.csv file within the zip file containing the Supplementary Materials.

**Acknowledgments:** We would like to thank Mihai Voicu, Tyler Rea, and Benjamin Paulsen for selecting the plot locations, measuring the trees and harvesting the trees used in this study, Nicole Wozney, Alex Lanti-Traikovski, and Lochlan Munro for weighing and recording the dry AGB, Jennifer Thomas for copyediting the original manuscript of this paper; and the Remote Sensing and Geographic Information Systems group at Northern Forestry Centre and the Centre for Earth Observation Sciences lab group at the University of Alberta for their feedback and support.

**Conflicts of Interest:** The authors declare no conflict of interest.

## Appendix A

*Uncertainty Propagation*

Final AGB estimates from the models are subject to uncertainties that stem from the model coefficients as well as from the instruments used to measure the variables being used as predictors in the model. These uncertainties can be seen as error bars on the model estimates in the predicted versus observed plots in Figures S1–S4. These uncertainties were calculated using error propagation formulas as outlined below. When predictors consisted of the product of two variables (i.e., $x = y \cdot z$ where $x$ is the predictor and $y$ and $z$ are the two variables used), measurement uncertainties for $y$ and $z$ were measured and combined using Equation (A1) for propagating errors through multiplication by maintaining the uncertainty percentages of each variable:

$$\delta x = |x| \sqrt{\left(\frac{\delta y}{y}\right)^2 + \left(\frac{\delta z}{z}\right)^2}. \tag{A1}$$

In the power models, which follow the form AGB $= \beta\, x^\alpha$, $\alpha$ and $\beta$ are model coefficients with standard error values of $\delta\alpha$ and $\delta\beta$. The uncertainty of the AGB estimates is represented by Equation (A1), where $y$ is replaced by $\beta$, $\delta y$ is replaced by $\delta\beta$, $\delta z$ is replaced by $\delta x^\alpha$, and $z$ is replaced by $x^\alpha$. To do this, we need to determine the uncertainty of the term $x^\alpha$, which can be undertaken by adding the linear approximations of $x$ and $\alpha$ in quadrature. The linear approximation of $x$ can be calculated as

$$\delta x^\alpha{}_1 = \alpha \cdot x^{\alpha-1} \cdot \delta x, \tag{A2}$$

and the linear approximation of $\alpha$ can be calculated as

$$\delta x^\alpha{}_2 = x^\alpha \cdot ln(x) \cdot \delta\alpha. \tag{A3}$$

These are then added together as

$$\delta x^\alpha = \sqrt{(\delta x_1^\alpha)^2 + (\delta x_2^\alpha)^2} \tag{A4}$$

The same process can be performed for the multiple regression power models by expanding Equation (A1) to encompass a third term (Equation (A5)):

$$\delta x = |x| \sqrt{\left(\frac{\delta m}{m}\right)^2 + \left(\frac{\delta y}{y}\right)^2 + \left(\frac{\delta z}{z}\right)^2}. \tag{A5}$$

Both terms with variables and coefficients are calculated as described using Equations (A2)–(A4).

Quadratic models follow a similar concept where we compute the uncertainty of each term from the model equation that takes the form

$$\text{AGB} = exp(\alpha x^2 + \omega x + \beta), \tag{A6}$$

where $\alpha$, $\omega$ and $\beta$ are model coefficients and $x$ is the predictor variable. The uncertainty in the first term, $\alpha x^2$, comes from the model uncertainty in $\alpha$ ($\delta\alpha$) and the measurement uncer-

tainty in $x^2$ ($\delta x^2$). The measurement uncertainty in $x^2$ can be determined using the linear approximation of $x^2$ and the measurement uncertainty of $x$ ($\delta x$) as shown in Equation (A7):

$$\delta x^2 = 2x \cdot \delta x. \tag{A7}$$

The $\delta x^2$ and $\delta \alpha$ terms can then be combined in the same fashion as outlined in Equation (A1) to obtain $\delta \alpha x^2$. Similarly, the $\delta \omega$ and $\delta x$ uncertainties that apply to the second term can be combined in the same way. These uncertainties are then added in quadrature (Equation (A8)) to obtain the total uncertainty for the measurements inside the brackets of Equation (A6).

$$\delta T = \sqrt{\left(\delta \alpha x^2\right)^2 + \left(\delta \omega x\right)^2 + \left(\delta \beta\right)^2}. \tag{A8}$$

Finally, the uncertainty of the AGB estimates can be found using Equation (A9):

$$\delta \mathrm{AGB} = \mathrm{AGB} \cdot \delta T. \tag{A9}$$

More details on error propagation formulas can be found in An Introduction to Error Analysis: The Study of Uncertainties in Physical Measurements [88].

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
