# Peer review of "Using TLS-Measured Tree Attributes to Estimate Aboveground Biomass in Small Black Spruce Trees"

_forests, doi:10.3390/f12111521_

Round 1
Reviewer 1 Report
Detailed comments:
Abstract
- L6: “Trees remove carbon from the atmosphere and store it as biomas” – Well said! Sometimes is good to remember the basics!
- L18: ” We fitted power, 18 quadratic, and multiple regression ... ” is it relevant? The abstract is really good as it is although it doesn´t hurt to remove the details on the modelling part. But here it goes my concern: TLS has been widely used and this is another study: what is the new stuff this study is bringing? What is the role of peatlands here? Are trees growing over peatlands in Canada very remarkable in terms of tree allometry? It is important to highlight the novelty of this work as there are hundreds papers already out there providing allometrics derived with TLS. Should we put this manuscript in that bag? Or is this something out of the TLS blue?
- L23: “rapid ground estimation” – not sure if TLS is a rapid technology. I can buy that for handheld TLS where transects and efficient path configurations can make the technology feasible to switch from plot-level allometric equattions to landscape/compartment level. The rapid term does not really apply for TLS thinking towards prediction out of the sampling design as tree positions and heights are hard to get with publicly available ALS datasets. Only with high-dense LiDAR (drone-based usually) the TLS-based models can be rapidly extrapolated to new point cloud-based datasets.
Introduction
- L28-40: So many lines to bring the point of climate change and carbon cycle. Shorten, please.
- L41: You guys can start with this – good paragraph.
- L60: ALS arrived earlier than TLS -it would be fair to say some words about the aerial side of laser scanning maybe.
- L65-80: Main limitation is data coverage to solve landscape inventory. Need some fair words on the suitability of TLS for plot-level purposes (allometries) but the restrictions on data coverage and time and wall-to-wall products.
- L81-88: Good point and I can see why canopies are more attractive in TLS mapping compared to DBH especially when thinking towards data fusion with other forms of scanning, mainly ALS and UAV LiDAR. The problem though is that there is a rich background on allometric modelling from the past linking always DBH with stages of development, usually in terms of height. The exercise is interesting and if this is the main novelty/purpose of the study this must be clearly highlight and “better sold”.
- L81-L83: Agree on biomass, do not agree on height which is sometimes used as intermediate step: DBH-height-all rest. Like the idea of trying to estimate AGB directly, at least try and see what can be obtained. Important to recognize previous works here and polish/highlight/better pitch the new study in terms of originality/novelty & need.
- L89-L93: This work can be done for peatlands, but also for other types of ecosystems. What the estimation of AGB ignoring DBH is more meaningful in peatlands ecosystems? Peatlands are important and research might have under-represented peatlands in previous studies. However, what the authors are trying here looks independent of the type of forest ecosystem. Just some thoughts here. The need to monitor peatlands is out of question, and that´s fair to say.
- L106-L116: Great last paragraph.
- In the Introduction I miss some few words about the expansion of the analyses or benchmark with other technologies. ALS is mentioned, correct. But if the purpose is to predict AGB wall-to-wall across the gigant, vast extension of peatlands in Canada, it seems reasonable to bring GEDI Laser Scanning mission product Level 4 here - recently released. See the work from Duncanson, Armstrong, Dubayah and many others on GEDI-based AGB estimates. Lot of studies comparing ALS versus GEDI.
Material and Methods
- Sections 2.1 and 2.2 are phenomenal – well written and very descriptive. Congrats
- Section 2.3 – brief but complete description of the field work. Love the addition of the GoPro for a cool monitoring and register of datasets.
- Section 2.4 – lot of groundwork that hopefully TLS and remote sensing data products help to overcome so we stop chopping down trees for destructive sampling!
- Section 2.4 – Could you provide a range/summary table/histogram of the final computed real volumes?
- Section 2.5 – listing the commands/specific routines in CloudCompare would be nice.
- Section 2.8 – see previous comment on the Cloud Compare software.
- The presence of small outliers could impact the volume computations in the way authors report. Question: Were identified trees and allometries visually inspected to detect anomalies along the ever-decreasing DBH with increasing height? Trunk areas close to neighbors or intersecting the branches of adjacent trees could lower the number of points in those sections, resulting in irregularities on the expected pattern of the allometry.
- Section 2.9 - The list of predictor variables does not look elegant in that way, use a symbol and describe later. The bullet list is not needed, just a paragraph with the symbol and spell out to properly explain the variable.
- Section 2.10 does not look as good, and I need some clarifications: authors are trying to create the raster ALS would do but using TLS technology. The raster format is not something exclusive and unique of the ALS technologies, point cloud statistics are also used and not raster-based. Canada has many, many areas with public ALS data. Have the authors checked this? If ALS exists to the study area, authors should try to create DTMs, CHMs and then perform the analyses indicated. The sensing capabilities in ALS and TLS are different because of point density but also for the nature of the scanning itself: TLS for DBH and ALS for height - this is a standard reasoning when it comes to scanning and that´s why TLS is used for allometries and we predict DBH from ALS-based heights. Can you guys check and process open ALS data? The size of trees scanned here is very small, if ALS is not high-density then DTM and some tree tops would be close to merge – height-break filter to disregard shrub from trees might not exist. Challenging with ALS at tree level.
- L302: Remove bullet points – same stylish aspect as the previous comment.
- L310-313: That´s why is important to report earlier the number of sampled trees by number & age class & diameter. To see whether this filter leaves out an important part of the experiment.
- Section 2.10:
Results
- Maybe Table 3 is part of the methods and not really thee results.
- Figure 2. Adding an RGB shot of the tree would be great
- Figure 3 is what I was asking. It might make more sense to include it the methods so we have a clear picture of the sort of trees we are talking here.
- Figure 3 can actually be a multi-part figure adding DBH and tree height in the X-axis and AGB in the Y-axis.
- Figure is wrong. Authors cannot report results on the caption.
- Has WLS been spelled out? The similarity with TLS and ALS advocated for not use an acronym and then use weighted linear squares as such
- Table 5 and 6 recombine or just use text to report
- Figure 6 is great. Check resolution – not good - in Figure 7 and maybe use b&w for scaling instead of color.
- Figure 8 left could be more informative using relative RMSE (%)
- The section is quite long and I feel some of the figures can be omitted such as Figure 5 – cool though. Same as Tables 4-5-6 all consecutive and of low dimension. Could we just help the reader to focus on the important?
- Line 413-418 - Formatting
Discussion
- To be checked in the next round. Still some important additions in the methods and results are needed before evaluating the discussion.
Author Response
[Authors] Thank you for your thorough and detailed feedback. We appreciate you taking the time to help us to improve our manuscript. Below is a summary of the changes made and a response to each of the points made in your review.
L6: “Trees remove carbon from the atmosphere and store it as biomas” – Well said! Sometimes is good to remember the basics!
Thank you!
” We fitted power, 18 quadratic, and multiple regression ... ” is it relevant? The abstract is really good as it is although it doesn´t hurt to remove the details on the modelling part. But here it goes my concern: TLS has been widely used and this is another study: what is the new stuff this study is bringing? What is the role of peatlands here? Are trees growing over peatlands in Canada very remarkable in terms of tree allometry? It is important to highlight the novelty of this work as there are hundreds papers already out there providing allometrics derived with TLS. Should we put this manuscript in that bag? Or is this something out of the TLS blue?
[Authors] We added a sentence to explain that black spruce in peatlands grow differently than those in upland forests and are generally stunted in their heights. We removed mention to the kinds of models, but we think it is still important to mention the predictors used to show the diligence done in trying to find alternatives to conventional DBH/Height models.
- L23: “rapid ground estimation” – not sure if TLS is a rapid technology. I can buy that for handheld TLS where transects and efficient path configurations can make the technology feasible to switch from plot-level allometric equattions to landscape/compartment level. The rapid term does not really apply for TLS thinking towards prediction out of the sampling design as tree positions and heights are hard to get with publicly available ALS datasets. Only with high-dense LiDAR (drone-based usually) the TLS-based models can be rapidly extrapolated to new point cloud-based datasets.
[Authors] We have completely reworded this sentence and it now reads:” Our equations are based on predictors that can be measured from above, therefore they may enable the plotless creation of accurate biomass reference data for a prominent tree species in the most common ecosystem (treed peatlands) of North America’s boreal”. ‘Rapid’ was referring to the potential to completely bypass field plots that have to be TLSed or measured from the ground.
- L28-40: So many lines to bring the point of climate change and carbon cycle. Shorten, please.
[Authors] We’ve removed the first paragraph as you recommend and begin with L41.
- L60: ALS arrived earlier than TLS -it would be fair to say some words about the aerial side of laser scanning maybe.
[Authors] We added a sentence to introduce ALS and its main usage in a forestry context.
- L65-80: Main limitation is data coverage to solve landscape inventory. Need some fair words on the suitability of TLS for plot-level purposes (allometries) but the restrictions on data coverage and time and wall-to-wall products.
[Authors] We added a couple sentences to note the limitations of TLS.
- L81-88: Good point and I can see why canopies are more attractive in TLS mapping compared to DBH especially when thinking towards data fusion with other forms of scanning, mainly ALS and UAV LiDAR. The problem though is that there is a rich background on allometric modelling from the past linking always DBH with stages of development, usually in terms of height. The exercise is interesting and if this is the main novelty/purpose of the study this must be clearly highlight and “better sold”.
[Authors] We changed this section to note examples of past studies done to develop biomass equations using DBH from areas around the world to highlight the wide-ranging history of these models. Also added some information to better explain the reason why we would want allometric equations based on crown parameters rather than DBH as has been more common in the past.
- L81-L83: Agree on biomass, do not agree on height which is sometimes used as intermediate step: DBH-height-all rest. Like the idea of trying to estimate AGB directly, at least try and see what can be obtained. Important to recognize previous works here and polish/highlight/better pitch the new study in terms of originality/novelty & need.
[Authors] We have changed the wording at the beginning of the sentence to focus the point on allometric biomass equations, which are the subject of this study (not allometric equations to estimate height using DBH).
- L89-L93: This work can be done for peatlands, but also for other types of ecosystems. What the estimation of AGB ignoring DBH is more meaningful in peatlands ecosystems? Peatlands are important and research might have under-represented peatlands in previous studies. However, what the authors are trying here looks independent of the type of forest ecosystem. Just some thoughts here. The need to monitor peatlands is out of question, and that´s fair to say.
[Authors] Individual tree AGB models are often species specific, and ours are only validated for small (< 5m ) black spruce trees, as there were hardly any trees taller than that in our study. Our models could perhaps be applicable to young upland black spruce (this would need to be tested though, as their architecture might be different than in peatlands), but those young trees do not remain shorter than 5 m for long, whereas in peatlands they may remain that short for their entire lives, hence the focus on peatlands. There are very few forest inventory (FI) plots in peatlands, so the aerial, plotless creation of AGB reference data through models not reliant on DBH is more relevant here than in the commercial forest where FI plots are more common.
- L106-L116: Great last paragraph.
[Authors] Thank you!
- In the Introduction I miss some few words about the expansion of the analyses or benchmark with other technologies. ALS is mentioned, correct. But if the purpose is to predict AGB wall-to-wall across the gigant, vast extension of peatlands in Canada, it seems reasonable to bring GEDI Laser Scanning mission product Level 4 here - recently released. See the work from Duncanson, Armstrong, Dubayah and many others on GEDI-based AGB estimates. Lot of studies comparing ALS versus GEDI.
[Authors] We now mention the need for extensive biomass reference data in order to upscale to satellite data (lines 90-98). The methods outlined here may allow for creating those reference data without necessitating ground estimates. In terms of GEDI, there is unfortunately no data available for the areas that we were studying as GEDI only covers to 51° and our study area was north of 60°. That said, we have added to the third last paragraph a note on GEDI and ICESat-2.
- Sections 2.1 and 2.2 are phenomenal – well written and very descriptive. Congrats
[Authors] Thank you!
- Section 2.3 – brief but complete description of the field work. Love the addition of the GoPro for a cool monitoring and register of datasets.
[Authors] Thanks, the GoPro images were quite helpful in registering the scans.
- Section 2.4 – lot of groundwork that hopefully TLS and remote sensing data products help to overcome so we stop chopping down trees for destructive sampling!
Section 2.4 – Could you provide a range/summary table/histogram of the final computed real volumes?
[Authors] We agree with the point on destructive sampling. Volume was not measured in the lab. We have moved Figure 3 up to this section (We think this is what you meant by your comments in the reference section).
- Section 2.5 – listing the commands/specific routines in CloudCompare would be nice.
[Authors] We added the name of the tool used in Cloud Compare, as well as the value we entered for the user parameters.
- Section 2.8 – see previous comment on the Cloud Compare software.
[Authors] Once the harvested trees are individually segmented into separate point clouds (described in 2.5), Cloud Compare automatically creates the bounding box for each tree and reports the box dimensions. We have added a sentence to clarify this.
- The presence of small outliers could impact the volume computations in the way authors report. Question: Were identified trees and allometries visually inspected to detect anomalies along the ever-decreasing DBH with increasing height? Trunk areas close to neighbors or intersecting the branches of adjacent trees could lower the number of points in those sections, resulting in irregularities on the expected pattern of the allometry.
[Authors] Yes, the QSMs and individual tree point clouds were visually inspected to confirm the DBHs that were being returned. In many cases, these values did not match up well and were often overestimated due to issues of occlusion and other branches near to the stem that were incorrectly interpreted as part of the stem. Part of the issue there was that there were often returns from the needles as well that made it difficult for the QSM script to delineate individual branches and stems (particularly with small trees). This is discussed further in section 4.2 - QSM Effectiveness.
- Section 2.9 - The list of predictor variables does not look elegant in that way, use a symbol and describe later. The bullet list is not needed, just a paragraph with the symbol and spell out to properly explain the variable.
[Authors] We have changed the bullet list to follow your suggestion.
- Section 2.10 does not look as good, and I need some clarifications: authors are trying to create the raster ALS would do but using TLS technology. The raster format is not something exclusive and unique of the ALS technologies, point cloud statistics are also used and not raster-based. Canada has many, many areas with public ALS data. Have the authors checked this? If ALS exists to the study area, authors should try to create DTMs, CHMs and then perform the analyses indicated. The sensing capabilities in ALS and TLS are different because of point density but also for the nature of the scanning itself: TLS for DBH and ALS for height - this is a standard reasoning when it comes to scanning and that´s why TLS is used for allometries and we predict DBH from ALS-based heights. Can you guys check and process open ALS data? The size of trees scanned here is very small, if ALS is not high-density then DTM and some tree tops would be close to merge – height-break filter to disregard shrub from trees might not exist. Challenging with ALS at tree level.
[Authors] Unfortunately, no publicly available ALS datasets exist for our study area. Even if they existed, ALS generally provides point densities of less than 10 pts/m2, so most likely we couldn’t do this analysis with conventional ALS data, as the reviewer reckons at the end of this comment, and as the study by Peng et al (2021) corroborates (they found that the accuracy of individual tree height predictions decreases rapidly once point densities fall below 17 pts/m2). The simulated experiment in section 2.10 was performed to gain some insight into where this threshold could lie for crown area measurements. We are currently working on an experiment with real data that tests the accuracy of these measurements (of both height and crown area) with multiple point clouds of varying point densities from both drones and planes.
- L302: Remove bullet points – same stylish aspect as the previous comment.
[Authors] We have removed this bullet list following your suggestion, thanks.
- L310-313: That´s why is important to report earlier the number of sampled trees by number & age class & diameter. To see whether this filter leaves out an important part of the experiment.
[Authors] While a relevant variable, the age of trees was not calculated as part of this study. The new figure 2 (multipart figure with height vs AGB and DBH vs AGB) in section 2.4, suggested by the reviewer as a modification of Figure 3 in the original manuscript), now gives a good overview of the sizes of trees being used in this study. Additionally, readers can find more information in table 2 of the Results section.
- Maybe Table 3 is part of the methods and not really thee results.
[Authors] Figure 3 has been moved to the methods (section 2.4). (We think this is what you were referring to with this note, because Table 3 shows the results for MAE, adj R2 and RMSE, so it makes sense for it to be in the results).
- Figure 2. Adding an RGB shot of the tree would be great
[Authors] We have added an RGB image of the unoccluded tree point cloud to this figure.
- Figure 3 is what I was asking. It might make more sense to include it the methods so we have a clear picture of the sort of trees we are talking here.
[Authors] We moved Figure 3 to the methods (see note above).
- Figure 3 can actually be a multi-part figure adding DBH and tree height in the X-axis and AGB in the Y-axis.
[Authors] We removed the histogram, and created a 2-part figure (now Fig 2) that shows a scatter plot of DBH vs AGB and height vs AGB instead (see comment for L310-313).
- Figure is wrong. Authors cannot report results on the caption.
[Authors] Figure 4. (We think this is the figure you were saying was incorrect?) We removed the information on the R2s and RMSEs that resulted from those models and reworded the caption to show it is a comparison between the weighted and ordinary least square methods.
- Has WLS been spelled out? The similarity with TLS and ALS advocated for not use an acronym and then use weighted linear squares as such
[Authors] WLS and OLS were spelled out earlier in the article (L296 when markup is turned off) and the same is true of TLS and ALS (L49-50). We’ve reworded the relevant sentences to remove any ambiguity.
- Figure 6 is great. Check resolution – not good - in Figure 7 and maybe use b&w for scaling instead of color.
[Authors] We checked the resolutions of both and it was 300 dpi, but we decided to recreate and upload them anyways in case something odd happened along the process somewhere. We also made Figure 7 grey scale instead of colour.
- Figure 8 left could be more informative using relative RMSE (%)
[Authors] We changed RMSE to relative RMSE per your suggestion.
- The section is quite long and I feel some of the figures can be omitted such as Figure 5 – cool though. Same as Tables 4-5-6 all consecutive and of low dimension. Could we just help the reader to focus on the important?
[Authors] We feel this figure provides valuable information within the context of the manuscript because it provides a good visualization for the best model that we created in this study. It allows the reader to see how the estimation plane for the model lies very close to the field measured data points. For that reason we feel it is best to leave Figure 5 in. Likewise, we think it is important to keep the information for the coefficients of the equations in the main manuscript, because the equations are the main deliverable from the paper. We have consolidated all this information into one table to help reduce some of the clutter.
- Line 413-418 - Formatting
[Authors] These lines are a part of the main text, we’ve also moved this paragraph up above Tables 3 and 4, and put Figure 5 below the tables to hopefully make it a bit clearer now.

Reviewer 2 Report
Wagers et al. used terrestrial LiDAR (TLS) data to developed biomass allometry models for the Black spruce, a dominant species in Canadian peatlands – an ecosystem type that cover a large proportion of the country. The authors notably focused on models that could be used with airborne LiDAR data, which has interesting practical implications. Models were developed on 100 destructively sampled trees from 10 inventory plots. The analysis is thorough. Overall, I enjoyed reading this study, which I find well-structured and well-written. I particularly enjoy the exhaustive description of the methodology, as well as the transparence / discussion points on the limits of the study and TLS data in general, which is quite refreshing. I only have a few minor comments.
462-464 : As a reader, after reading this sentence I wonder what’s the size distribution of tree crowns in your dataset. I went back to Tab 2 and this information is not there. Maybe it would be worth adding it (like average +/- sd by plot) ? In my opinion, it’s more important than, e.g. “# of Black Spruce in Plot”.
496-497 : Indeed. Note that in most studies where TLS is used to develop AGB models, the scanning protocol is “tree-centered”, in that several scans are made focusing on a single focal tree (the one that will be destructively sampled), trees around this focal tree are removed, etc…, as opposed to your ‘plot-centered’ scanning protocol which is increasingly becoming the norm. This has important consequences on oncclusion, thus on the quality of QSMs. An interesting study on this issue is :
Martin-Ducup, O., Mofack, G., Wang, D., Raumonen, P., Ploton, P., Sonké, B., ... & Pélissier, R. (2021). Evaluation of automated pipelines for tree and plot metric estimation from TLS data in tropical forest areas. Annals of Botany.
541-545 : I should note that when I looked at Fig. 1 my first thought was “plots are heavily clustered”. We can easily visualize 4 spatial clusters / sampling sites. You decided to use leave-one-plot-out validation rather than a simple leave-one-out (or random 10-fold) cross-validation as you acknowledge that there is a potential underlying correlation between trees within plots (so that entire plots should be remove from model training, and kept for testing). However, the spatial range of this underlying correlation is unknown. It could extend far beyond plot size, and neighboring plots could have been grouped into “sampling sites” to perform a leave-one-site-out validation. I’m not saying that it is what you should do, but it would worth checking if the results are stable with such cross-validation.
Author Response
Wagers et al. used terrestrial LiDAR (TLS) data to developed biomass allometry models for the Black spruce, a dominant species in Canadian peatlands – an ecosystem type that cover a large proportion of the country. The authors notably focused on models that could be used with airborne LiDAR data, which has interesting practical implications. Models were developed on 100 destructively sampled trees from 10 inventory plots. The analysis is thorough. Overall, I enjoyed reading this study, which I find well-structured and well-written. I particularly enjoy the exhaustive description of the methodology, as well as the transparence / discussion points on the limits of the study and TLS data in general, which is quite refreshing. I only have a few minor comments.
[Authors] Thank you for your kind words on the manuscript and the feedback you provided. We appreciate you taking the time to help us to improve it. Below is a response with the changes that were made or a response to the specific points that you made.
462-464 : As a reader, after reading this sentence I wonder what’s the size distribution of tree crowns in your dataset. I went back to Tab 2 and this information is not there. Maybe it would be worth adding it (like average +/- sd by plot) ? In my opinion, it’s more important than, e.g. “# of Black Spruce in Plot”.
[Authors] We have removed the column with number of trees used per plot and replaced it with average crown area, standard deviation and range (per plot and all combined). Because there were other variables used in this study, we have also included a table in the supplementary materials that contains the average, standard deviation and ranges for each of those variables.
496-497 : Indeed. Note that in most studies where TLS is used to develop AGB models, the scanning protocol is “tree-centered”, in that several scans are made focusing on a single focal tree (the one that will be destructively sampled), trees around this focal tree are removed, etc…, as opposed to your ‘plot-centered’ scanning protocol which is increasingly becoming the norm. This has important consequences on oncclusion, thus on the quality of QSMs. An interesting study on this issue is :
Martin-Ducup, O., Mofack, G., Wang, D., Raumonen, P., Ploton, P., Sonké, B., ... & Pélissier, R. (2021). Evaluation of automated pipelines for tree and plot metric estimation from TLS data in tropical forest areas. Annals of Botany.
[Authors] Thanks, that is an interesting study and so is the idea of a plot-centered approach having a major effect on the occlusion. While tree-centered scans would have virtually eliminated any occlusion, it just wouldn’t be feasible in dense treed peatlands like these, as the time needed to clear out all surrounding trees and take multiple scans for each tree would have increased the time and cost of data collection drastically or resulted in fewer trees being scanned. We have added a sentence in the discussion (S4.2) to mention that the plot centered scanning approach may have also played a role in the ineffectiveness of the QSMs for measuring DBH and volume.
541-545 : I should note that when I looked at Fig. 1 my first thought was “plots are heavily clustered”. We can easily visualize 4 spatial clusters / sampling sites. You decided to use leave-one-plot-out validation rather than a simple leave-one-out (or random 10-fold) cross-validation as you acknowledge that there is a potential underlying correlation between trees within plots (so that entire plots should be remove from model training, and kept for testing). However, the spatial range of this underlying correlation is unknown. It could extend far beyond plot size, and neighboring plots could have been grouped into “sampling sites” to perform a leave-one-site-out validation. I’m not saying that it is what you should do, but it would worth checking if the results are stable with such cross-validation.
[Authors] There were no formal clusters defined in our study, but the reviewer is right that the locations are not evenly distributed. The plots, however, are kilometers apart from the closet plots, so the likelihood that trees from neighbouring plots are within the range of spatial autocorrelation of the response or predictor variables is low. That said, we have now tried a ‘leave one cluster out’ (LOCO) cross-validation out of curiosity (4 clusters, two of 3 plots and two of 2) and the results are the following: Mean Tree AGB Error (kg): [0.253, 0.284, 0.228, 0.159]; individual tree AGB RMSE (kg): [0.328, 0.622, 0.377, 0.229] The 0.622 RMSE seems to be caused by a single tree in plot #12 that had an odd shape (in fact the RMSE when that plot is left out under LOPO is even larger, 0.88 kg). Otherwise results seem pretty stable, and in any case, the results from either LOCO or LOPO are similar (e.g., the mean RMSE in LOCO is 0.39kg, and in LOPO it is 0.35 kg, and their coefficients of variation are 44% and 43% respectively.
